# MaIL: Improving Imitation Learning with Selective State Space Models

**Xiaogang Jia** [*†‡] **Qian Wang**[*†] **Atalay Donat**[†] **Bowen Xing**[‡] **Ge Li**[†]
**Hongyi Zhou**[‡] **Onur Celik**[†] **Denis Blessing**[†] **Rudolf Lioutikov**[‡] **Gerhard Neumann**[†]
[†] Autonomous Learning Robots, Karlsruhe Institute of Technology
[‡] Intuitive Robots Lab, Karlsruhe Institute of Technology

**Abstract:**

This work presents **Ma**mba **I**mitation **L**earning (MaIL), a novel imitation learning (IL) architecture that provides an alternative to state-of-the-art (SoTA) Transformer-based policies. MaIL leverages Mamba, a state-space model designed to selectively focus on key features of the data. While Transformers are highly effective in data-rich environments due to their dense attention mechanisms, they can struggle with smaller datasets, often leading to overfitting or suboptimal representation learning. In contrast, Mamba's architecture enhances representation learning efficiency by focusing on key features and reducing model complexity. This approach mitigates overfitting and enhances generalization, even when working with limited data. Extensive evaluations on the LIBERO benchmark demonstrate that MaIL consistently outperforms Transformers on all LIBERO tasks with limited data and matches their performance when the full dataset is available. Additionally, MaIL's effectiveness is validated through its superior performance in three real robot experiments. Our code is available at https://github.com/ALRhub/MaIL.

**Keywords:** Imitation Learning, Sequence Models, Denoising Diffusion Policies

## 1 Introduction

Imitation learning (IL) [1] from human data has shown remarkable success in acquiring robot policies that can solve complex tasks [2, 3, 4, 5, 6]. As human behavior is inherently multi-modal and non-Markovian, prior research has demonstrated great benefits in using historical information [7], predicting a sequence of actions instead of a single action [3, 5] and using methods that can model multi-modal distributions [5, 6, 8]. Recent works [5, 6] therefore base their policies on Transformer models to effectively handle *sequences of observations*. The Transformer's self-attention and cross-attention mechanisms have led to remarkable results in various domains [9, 10, 11] and are considered state-of-the-art for processing sequential data. Here, current approaches either use an decoder-only structure [5], or a decoder-encoder architecture [6]. Which of these architectures excels is often task dependent. The performance of transformers usually comes with large models that are difficult to train, especially in domains where data is scarce. An alternative concept for handling a *sequence of observations* are state space models [12]. These models assume a linear relationship between observations (embeddings) and are usually computationally more efficient. Recent approaches such as Mamba [13], a selective state space model, rigorously improve the performance of state space models and rival against transformers in many tasks. Due to its properties in inference speed, memory usage, and efficiency, Mamba is an appealing model for IL policies.

This work proposes **MaIL**, a novel imitation learning policy architecture that uses Mamba as a backbone. MaIL can be used as a standalone policy, or as part of more advanced processes such as a

---

[*]Equal contribution, correspondence to jia266163@gmail.com

diffuser in the diffusion process. We implement MaIL in two variants. In the decoder-only variant, MaIL processes the noised actions and the observation features [5] together with the time embedding of the diffusion process and outputs the denoised actions. However, due to Mamba's formalism, an encoder-decoder variant is not straightforwardly implementable due to possibly varying input and output structures. We show how we can extend MaIL to an encoder-decoder variant by extending the inputs with learnable action, state, and time embedding variables such that learning can still be done efficiently. We show that this model works more efficiently for multi-modal inputs such as image and language input. Our extensive evaluations on the state-of-the-art LIBERO [14] IL benchmarks with and without history input show that MaIL achieves improved performance compared to similarly complex transformer-based IL policies. MaIL's high performance is further confirmed on three challenging real robot experiments.

## 2   Related Works

**Sequence Models.** In recent years, Transformers [15, 10, 16] have become the leading approach in handling long-term dependencies in sequential data. The self-attention mechanism in Transformers allows processing sequences in parallel, effectively addressing the limitations of RNN in sequential data processing [17, 18, 19, 20]. However, structured state space models [12, 21, 22, 13] provide an appealing alternative to Transformers. While transformers scale quadratically in the sequence length, structured state space models scale linearly [13]. Earlier approaches [12, 23, 24] rely on a convolutional formulation of the learnable matrices that allows training in parallel and only requires sequential processing during inference. However, this formulation requires time and input-independent state space matrices harming the performance. In contrast, recent works [13] rely on associative scans that also allow for parallel computations but additionally allow for input-dependent learnable matrices [13]. For a more detailed review, we refer the reader to [25]. This work leverages the state-of-the-art method Mamba [13] that showed competing performance compared to transformers. MaIL exploits the selective state space models present in Mamba and proposes a novel architecture for improved performance in imitation learning, especially in sequential inputs.

**Imitation Learning (IL).** Early imitation learning methods primarily focused on learning one-to-one mappings between state-action pairs. Despite demonstrating promising results in tasks like autonomous driving [26] and robot control [27, 8, 28], these methods overlooked the rich temporal information contained in history. Subsequent approaches incorporated RNNs to encode observation sequences [29, 30, 31, 32], demonstrating that the utilization of historical observations can enhance model performance. However, these methods suffer from the inherent limitations of RNN-based architectures, including restricted representation power with long-sequence modeling and slow training times due to their unsuitability for large-scale parallelization. With the raise of Transformers in NLP and CV domains, modern imitation learning methods have adopted Transformers as their backbone [33, 34, 35]. Transformers can model long sequences while maintaining training efficiency through parallelized sequence processing. This trend extends to IL with multi-modal sensory inputs [36, 37, 38, 39], where Transformers encode both image and language sequences. Recently, Diffusion Models has demonstrated superiority in imitation learning [5, 40, 6, 41, 39]. Due to their strong generalization ability and rich representation power to capture multimodal action distributions, they have become the SoTA in the imitation learning domain. Many of these models also utilize Transformers as policies, leveraging their rich representation capabilities [6, 41, 39]. Instead of relying on a transformer backbone, MaIL proposes a novel architecture that is based on Mamba [13]. In the evaluations, we show the benefits of MaIL, especially in scenarios with sequences of observation inputs over state-of-the-art methods.

## 3   Preliminaries

Here, we briefly review the basics of the Mamba model and explain the policy representations considered in this work.

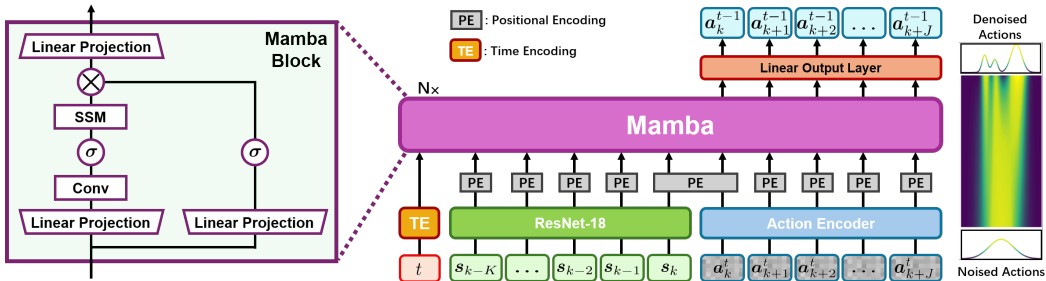

Figure 1: D-Ma: Mamba denoising architecture integrates ResNet-18 for state encoding and an action encoder for action encoding. The state sequence has a length of $K$, while the action sequence at diffusion step $t$ has a length of $J$. Before feeding the inputs into the Mamba module, positional encoding (PE) and time encoding (TE) enhance the inputs, where the $s_k$ and $a_k$ share the same positional encoding. The mamba module has N× mamba blocks, with a detailed structure [13] shown on the left. The outputs from the mamba module are processed by a linear output layer, resulting the one-step denoising actions. The symbol × in the Mamba block denotes matrix multiplication, and $\sigma$ the SiLU activation function.

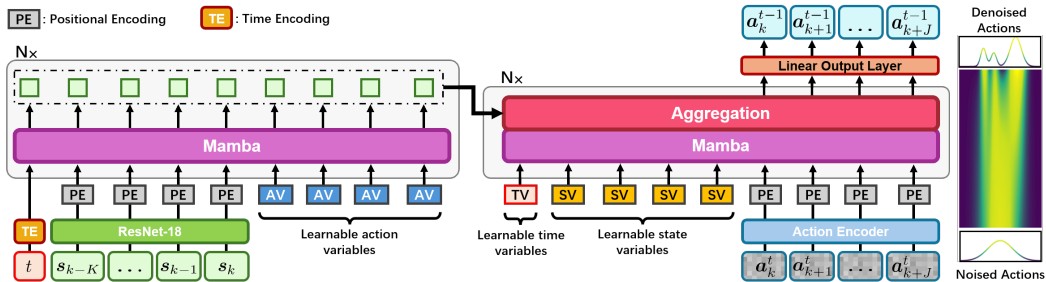

Figure 2: ED-Ma: Different from the D-Ma model, ED-Ma contains a Mamba encoder which is used to process the time embedding and state embedding, and a Mamba decoder which is used to process the noisy actions. In order to aggregate the information from encoder and decoder, learnable action variables are introduced to the encoder input and learnable time variables and state variables are introduced to the decoder output for sequence alignment.

### 3.1 Mamba: Selective State-Space Models

Inspired by the Attention mechanism [15] in Transformers, Mamba [13] improves upon the Structured State-Space Sequence Models (SSMs) [12] by using a selective scan operator, to propagate or forget information over time, allowing it to filter out relevant features. Specifically, we denote the input sequence $x \in \mathbb{R}^{B \times L \times D}$ and the output sequence $y \in \mathbb{R}^{B \times L \times D}$, where B, L, D refer to the batch size, sequence length and dimension respectively. A standard SSM with hidden state dimension N defines time-invariant parameters $\mathbf{A} \in \mathbb{R}^{N \times N}$, $\mathbf{B} \in \mathbb{R}^{N \times D}$, $\mathbf{C} \in \mathbb{R}^{D \times N}$ and the time-step vector $\mathbf{\Delta}$ to map the inputs up to $x(l)$ to a hidden state, which then can be projected to the output $y(l)$ (More details are in Appendix C.1). However, this property limits the SSM for contextual learning. Mamba implements the selection mechanism by making the SSM parameters a function of the input, that is,

$$\mathbf{B} = \text{Linear}(x), \ \mathbf{C} = \text{Linear}(x), \ \mathbf{\Delta} = \text{SoftPlus}(\text{Linear}(x)), \tag{1}$$

where $\mathbf{B} \in \mathbb{R}^{B \times L \times D}$, $\mathbf{C} \in \mathbb{R}^{B \times L \times D}$, and $\mathbf{\Delta} \in \mathbb{R}^{B \times L \times D}$, Linear refers to linear projection layers and SoftPlus is a smooth approximation of ReLU. Then the output can be calculated via

$$y = \text{SSM}(\overline{\mathbf{A}}, \overline{\mathbf{B}}, \mathbf{C})(x), \tag{2}$$

where $(\overline{\mathbf{A}}, \overline{\mathbf{B}})$ are discretized [42] counterparts of $(\mathbf{A}, \mathbf{B})$ with time-steps $\mathbf{\Delta}$. As the time-varying model can only be calculated in a recurrent way, Mamba further implements a hardware-aware method to compute the selective SSM efficiently. Figure 1 shows a simplified Mamba block.

| **Algorithm 1** D-Ma DDP Inference | **Algorithm 2** ED-Ma DDP Inference |
|---|---|
| **Require:** Observation $\mathbf{s}_{k-K:k}$, Mamba Decoder $\mathbf{Dec_m}$, Action Encoder $\mathbf{Enc_a}$, Position Embedding $\mathbf{PE}$ | **Require:** Observation $\mathbf{s}_{k-K:k}$, Mamba Encoder $\mathbf{Enc_m}$, Mamba Decoder $\mathbf{Dec_m}$, Position Embedding $\mathbf{PE}$, Learnable action variables $\hat{\mathbf{a}}_{k:k+J}$, learnable state variables $\hat{\mathbf{s}}_{k-K:k}$, learnable time variables $\hat{t}$ |
| 1: $\mathbf{a}_{k:k+J}^{T} \sim \mathcal{N}(\mathbf{0}, \mathbf{I})$ | 1: $\mathbf{a}_{k:k+J}^{T} \sim \mathcal{N}(\mathbf{0}, \mathbf{I})$ |
| 2: **for** $t = T, \dots, 1$ **do** | 2: **for** $t = T, \dots, 1$ **do** |
| 3: $\quad E_t = \mathbf{TE}(t)$ | 3: $\quad E_t = \mathbf{Linear}(t)$ |
| 4: $\quad E_{\mathbf{s}} = \mathbf{PE}(\mathbf{ResNet}(\mathbf{s}_{k-K:k}))$ | 4: $\quad E_s = \mathbf{PE}(\mathbf{ResNet}(\mathbf{s}_{k-K:k}))$ |
| 5: $\quad E_{\mathbf{a}} = \mathbf{PE}(\mathbf{Enc_a}(\mathbf{a}_{k:k+J}^{t}))$ | 5: $\quad E_a = \mathbf{PE}(\mathbf{Linear}(\mathbf{a}_{k:k+J}^{t}))$ |
| 6: $\quad \mathbf{a}_{k:k+J}^{t-1} =$ $\quad\quad \mathbf{Linear}(\mathbf{Dec_m}(E_t, E_{\mathbf{s}}, E_{\mathbf{a}}))$ | 6: $\quad E_s = \mathbf{E_m}(\text{cat}(E_t, E_s, \hat{\mathbf{a}}_{k:k+J}))$ $\quad\quad \triangleright$ Sequence alignment |
|  | 7: $\quad E_a = \mathbf{D_m}(\text{cat}(\hat{t}, \hat{\mathbf{s}}_{k-K:k}, E_a))$ $\quad\quad \triangleright$ First decoder layer |
| 7: **end for** | 8: $\quad \mathbf{a}_{k:k+J}^{t-1} = \mathbf{D_m}(E_s + E_a)$ $\quad \triangleright$ Remaining decoder layers |
| 8: **return** $\mathbf{a}_{k:k+J}$ | 9: **end for** |
|  | 10: **return** $\mathbf{a}_{k:k+J}$ |

## 3.2 Policy Representations

In this work, we use two policy representations: behavioral cloning (BC) and denoising diffusion policies (DDPs). For clarity, we focus on the non-sequential case.

**Behavioral Cloning** assumes a parameterized conditional Gaussian distribution as policy representation, i.e., $\pi(\mathbf{a}|\mathbf{s}) = \mathcal{N}(\mathbf{a}|\mu_\theta(\mathbf{s}), \sigma^2 \mathbf{I})$. Maximizing the likelihood for the model parameters $\theta$ simplifies to a mean-square error (MSE) loss, that is,

$$\mathcal{L}_{\mathrm{BC}}(\theta) = \mathbb{E}_{\mathbf{s}, \mathbf{a}} \left[ \|\mu_\theta(\mathbf{s}) - \mathbf{a}\|_2^2 \right], \tag{3}$$

where the expectation over $\mathbf{s}, \mathbf{a}$ is approximated using state-action pairs in the demonstration data.

**Denoising Diffusion Policies** utilize a denoising function $\epsilon_\theta$ to sample from a Markov chain $(\mathbf{a}^t)_{t=1}^T$

$$\mathbf{a}^{t-1} = \frac{1}{\sqrt{\alpha_t}} \left( \mathbf{a}^t - \frac{1 - \alpha_t}{\sqrt{1 - \bar{\alpha}_t}} \epsilon_\theta(\mathbf{a}^t, t, \mathbf{s}) \right) + \sqrt{\alpha_t} \mathbf{z}^t, \quad \text{where} \quad \mathbf{z}^t \sim \mathcal{N}(\mathbf{0}, \mathbf{I}), \tag{4}$$

starting from $\mathbf{a}_N \sim \mathcal{N}(\mathbf{0}, \mathbf{I})$ to produce a noise-free action $\mathbf{a}^0$ for a given observation $\mathbf{s}$. The denoising function is trained to predict the source noise $\mathbf{z}^0 \sim \mathcal{N}(\mathbf{0}, \mathbf{I})$ of a noisy action $\mathbf{a}^t = \sqrt{\bar{\alpha}_t} \mathbf{a}^0 + \mathbf{z}^t$ by minimizing the loss

$$\mathcal{L}_{\mathrm{DPM}}(\theta) = \mathbb{E}_{\mathbf{s}, \mathbf{a}, t} \left[ \|\epsilon_\theta(\mathbf{a}^t, t, \mathbf{s}) - \mathbf{z}^0\|_2^2 \right], \tag{5}$$

where $\bar{\alpha}_t = \prod_{j=1}^t \alpha_j$. The expectation over $t$ corresponds to a uniform sampling in $\{1, \dots, T\}$.

# 4 Mamba for Imitation Learning

In this section, we explain how we leverage Mamba for Imitation Learning (IL). Drawing inspiration from the successful Decoder-only (**D-Tr**) and Encoder-Decoder (**ED-Tr**) Transformers, we propose two Mamba-based architectures: Decoder-only Mamba (**D-Ma**) and Encoder-Decoder Mamba (**ED-Ma**). These architectures serve as the parameterization for the policy. Specifically, when employing Behavioral Cloning (BC), these architectures parameterize the mean $\mu_\theta$ of a conditional Gaussian distribution. Conversely, when using Denoising Diffusion Policies (DDPs), these architectures parameterize the denoising function $\epsilon_\theta$. Given the straightforward nature of the former case, we focus on introducing these architectures in the context of DDPs.

## 4.1 Decoder-Only Mamba

Similar to the Decoder-only transformer, we use a Mamba block to process the inputs. An overview of the Decoder-only Mamba architecture is shown in Figure 1. The Decoder-only Mamba for DDPs is designed to learn a denoising function $\epsilon_\theta$ that takes a sequence of observations $\mathbf{s}_{k-K:k}$, noisy

actions $\mathbf{a}_{k:k+J}^{t}$ and diffusion step $t$ to generate a less noisy sequence of actions $\mathbf{a}_{k:k+J}^{t-1}$. The diffusion step is encoded using time-embedding $\mathbf{TE}$. The observations are encoded using a $\mathbf{ResNet}$-18, with shared weights across images from different time steps. An action encoder $\mathbf{ENC_a}$ is used to tokenize the noisy action inputs. Additionally, position embeddings $\mathbf{PE}$ are applied to both the observations and actions. The time embedding, state embedding, and action embedding will then be inputted into the Mamba Decoder $\mathbf{Dec}_m$. The Mamba Decoder is implemented by stacking multiple mamba blocks with residual connections and Layer Normalization. The full inference routine is shown in Algorithm 1.

## 4.2 Encoder-Decoder Mamba

Compared to the Decoder-only Transformer containing only self-attention mechanisms, the Encoder-Decoder Transformer with cross-attention is a more flexible and effective design to process complex input-output relationships, especially for cases where the input and output sequences differ in structure. However, Mamba does not provide such a mechanism to support the encoder-decoder structure since the target and the source share the same sequence length. Here, we propose a novel approach called Mamba Aggregation which is used to design the encoder-decoder version of Mamba. The visualization can be found in Figure 2. The Mamba Encoder $\mathbf{Enc_m}$ is used to process the time embedding and state embedding, and the Mamba Decoder $\mathbf{Dec_m}$ is used to process the noise embedding. Since the inputs to the $\mathbf{E_m}$ and $\mathbf{D_m}$ have different lengths of sequences, we propose to add learnable variables to complement each sequence. The method is outlined in Alg 2.

## 5 Experiments

We conducted extensive experiments on both simulation benchmarks and real robot setups to verify the effectiveness of MaIL in imitation learning. Our investigation focuses on the following key questions:

**Q1**) Can MaIL achieve comparable or superior performance to Transformers?
**Q2**) Can MaIL utilize multi-modal inputs, such as language instructions?
**Q3**) How effectively do MaIL handle sequential information in observations?

### 5.1 Baselines

In this work, we mainly aim to explore the potential of Mamba in visual imitation learning compared to Transformer structures. Therefore, our experiments contain four architectures: Decoder-only Transformer (**D-Tr**), Encoder-Decoder Transformer (**ED-Tr**), Decoder-only Mamba (**D-Ma**), Encoder-Decoder Mamba (**ED-Ma**). For a fair comparison, we use ResNet18 to encode visual inputs for each method. For tasks that use language instructions, we use the pre-trained CLIP model [43] to get the corresponding language embedding which is used in training and inference for all methods. Based on the above setting, we implement the following imitation learning policies:

**Behavior Cloning (BC)** We implement a vanilla behavior cloning policy trained with MSE loss with both Transformer and Mamba structures.

**Denoising Diffusion Policies (DDP)** Based on the same structures in BC, we further implement a diffusion policy using a discrete denoising process [44]. We use 16 diffusion time steps for training and sampling for each architecture.

### 5.2 Simulation Evaluation

**LIBERO** [14]: The evaluation is conducted using the LIBERO benchmark, which encompasses five distinct task suites: *LIBERO-Spatial*, *LIBERO-Object*, *LIBERO-Goal*, *LIBERO-Long*, and *LIBERO-90*. Each task suite comprises 10 tasks along with 50 human demonstrations except for *LIBERO-90* which contains 90 tasks with 50 demonstrations. Each task suite is designed to test different aspects

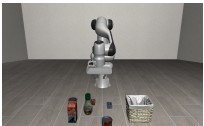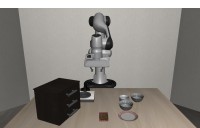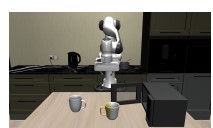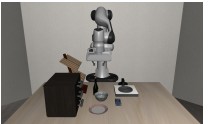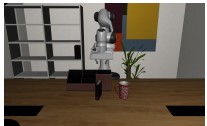

Figure 3: LIBERO benchmark suites with total 130 tasks in five different scenes.

| Policy | Backbone | w/o language | | | w/ language | |
| --- | --- | --- | --- | --- | --- | --- |
| | | LIBERO-Object | LIBERO-Spatial | LIBERO-Long | LIBERO-Goal | LIBERO-90 |
| BC-H1 | D-Tr | $0.358_{\pm0.086}$ | $0.257_{\pm0.030}$ | $0.222_{\pm0.067}$ | $0.362_{\pm0.034}$ | $0.311_{\pm0.055}$ |
| | ED-Tr | $0.480_{\pm0.027}$ | $0.333_{\pm0.029}$ | $0.265_{\pm0.015}$ | $0.325_{\pm0.020}$ | $0.268_{\pm0.050}$ |
| | D-Ma | $0.550_{\pm0.062}$ | $\mathbf{0.410_{\pm0.058}}$ | $\mathbf{0.287_{\pm0.037}}$ | $\mathbf{0.482_{\pm0.066}}$ | $\mathbf{0.559_{\pm0.021}}$ |
| | ED-Ma | $\mathbf{0.618_{\pm0.058}}$ | $0.352_{\pm0.069}$ | $0.275_{\pm0.023}$ | $0.473_{\pm0.031}$ | $0.514_{\pm0.008}$ |
| BC-H5 | D-Tr | $0.489_{\pm0.170}$ | $0.286_{\pm0.040}$ | $0.243_{\pm0.052}$ | $0.315_{\pm0.036}$ | $0.284_{\pm0.012}$ |
| | ED-Tr | $0.417_{\pm0.300}$ | $0.287_{\pm0.095}$ | $0.248_{\pm0.021}$ | $0.338_{\pm0.024}$ | $0.300_{\pm0.012}$ |
| | D-Ma | $\mathbf{0.765_{\pm0.115}}$ | $0.372_{\pm0.024}$ | $0.252_{\pm0.072}$ | $\mathbf{0.43_{\pm0.029}}$ | $0.529_{\pm0.094}$ |
| | ED-Ma | $0.743_{\pm0.020}$ | $\mathbf{0.428_{\pm0.029}}$ | $\mathbf{0.262_{\pm0.047}}$ | $0.317_{\pm0.105}$ | $\mathbf{0.599_{\pm0.019}}$ |
| DDP-H1 | D-Tr | $0.667_{\pm0.058}$ | $0.473_{\pm0.040}$ | $0.330_{\pm0.029}$ | $0.300_{\pm0.350}$ | $0.181_{\pm0.024}$ |
| | ED-Tr | $0.680_{\pm0.027}$ | $0.420_{\pm0.022}$ | $0.373_{\pm0.042}$ | $0.478_{\pm0.033}$ | $0.399_{\pm0.030}$ |
| | D-Ma | $\mathbf{0.728_{\pm0.032}}$ | $\mathbf{0.492_{\pm0.060}}$ | $0.343_{\pm0.059}$ | $0.510_{\pm0.043}$ | $0.405_{\pm0.139}$ |
| | ED-Ma | $0.713_{\pm0.008}$ | $0.483_{\pm0.031}$ | $\mathbf{0.395_{\pm0.035}}$ | $\mathbf{0.565_{\pm0.023}}$ | $\mathbf{0.603_{\pm0.017}}$ |
| DDP-H5 | D-Tr | $0.762_{\pm0.070}$ | $0.435_{\pm0.043}$ | $0.353_{\pm0.033}$ | $0.315_{\pm0.058}$ | $0.161_{\pm0.024}$ |
| | ED-Tr | $0.812_{\pm0.065}$ | $0.415_{\pm0.097}$ | $0.348_{\pm0.019}$ | $0.518_{\pm0.014}$ | $0.437_{\pm0.046}$ |
| | D-Ma | $\mathbf{0.815_{\pm0.004}}$ | $\mathbf{0.538_{\pm0.044}}$ | $0.387_{\pm0.040}$ | $0.388_{\pm0.027}$ | $0.352_{\pm0.096}$ |
| | ED-Ma | $0.778_{\pm0.005}$ | $0.513_{\pm0.024}$ | $\mathbf{0.417_{\pm0.023}}$ | $\mathbf{0.563_{\pm0.070}}$ | $\mathbf{0.455_{\pm0.081}}$ |

Table 1: Performance on LIBERO benchmark with 20% data, where "w/o language" indicates that we do not use language instructions and "w/ language" means we use language tokens generated from a pre-trained CLIP model, H1 and H5 refer to using current state and 5 steps historical states respectively.

of robotic learning and manipulation capabilities. The task visualizations are in Figure 3. More details can be found in Appendix B.

**Evaluation Protocol** We compared each method across five LIBERO task suites separately. Instead of using full demonstrations, we utilized only 20% of the demonstrations for each sub-task, amounting to 100 trajectories per task suite, except for LIBERO-90, which contains 900 trajectories. We tune the hyper-parameters for both Transformer and Mamba, making sure they have a similar amount of parameters. All models were trained for 50 epochs, and we used the last checkpoint for evaluation. Following the official LIBERO benchmark settings, we performed 20 rollouts for each sub-task, totaling 200 evaluations per task suite, except for LIBERO-90, which includes 1800 evaluations. We report the average success rate for each task suite over 3 seeds.

**Main Results**. We report the main results in Table 1. Our Mamba-based architectures, D-Ma and ED-Ma, significantly outperform Transformer-based methods across all LIBERO task suites based on the BC policy. Specifically, Mamba-based models achieve nearly a 30% improvement in success rate in LIBERO-Object and LIBERO-90. When using the DDP policy, our models consistently surpass the Transformer baselines, with performance improvements exceeding 5% in most tasks. These results confirm **Q1**, demonstrating that MaIL achieves superior performance compared to Transformers. To address **Q2**, we compare MaIL with Transformers on LIBERO-Goal and LIBERO-90, using additional language embeddings as inputs. We observe significant improvements with Mamba-based methods in these tasks, indicating that MaIL effectively leverages multi-modal inputs. Given that most recent visual imitation learning works use historical observations as inputs, we evaluated the methods with 1 and 5 historical observations. We found that historical information does not always enhance performance. Only in LIBERO-Object do the H5 models outperform the H1 models, while in other tasks, H5 models achieve similar or worse results. Mamba-based H5 models again perform consistently better than Transformer-based models, which indicates that MaIL is able to capture sequential observation features effectively, answering **Q3**.

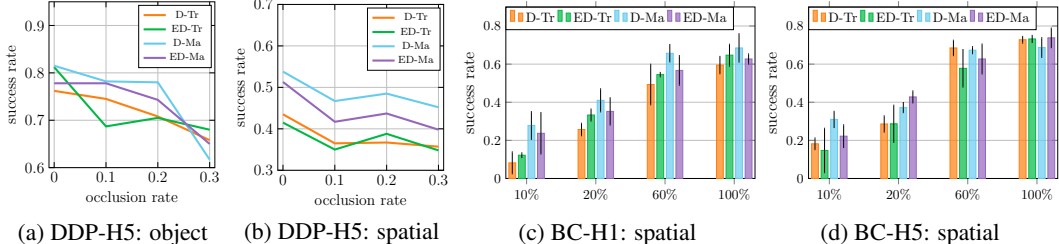

| (a) DDP-H5: object | (b) DDP-H5: spatial | (c) BC-H1: spatial | (d) BC-H5: spatial |

Figure 4: Ablation study for observation occlusions and dataset size. (a) and (b) study the influence of observation occlusions on Libero-Object and LIBERO-Spatial respectively. (c) and (d) show the influence of the amount of data on LIBERO-spatial.

**Ablation on Observation Occlusions** In order to further understand the sequential learning ability of Transformer and Mamba, we randomly mask out areas of images and test the model's performance drop. The results are reported in Figure 4. While for zero occlusions, the Transformer architectures can perform en par with Mamba, adding occlusions degenerates the performance of transformers more quickly, indicating that Mamba can better extract the important information out of the history sequence.

**Ablation on Dataset Size** Given that MaIL performs well with only 20% of the demonstrations, we are interested in evaluating its scalability with increasing dataset size. We compared Mamba-based models with Transformer models using the BC policy on the LIBERO-Spatial task. The results are presented in Figure 4. More experimental results can be found in Appendix Table 6. It is evident that Mamba-based models significantly outperform Transformers when data is scarce and perform comparably as the dataset size increases.

## 5.3 Real Robot Evaluation

We designed three challenging tasks based on the 7DoF Franka Panda Robot, utilizing visual inputs for the model. Two cameras, positioned at different angles in front of the robot, provide the visual data. One image is cropped and resized to $(128, 256, 3)$, while the other is resized to $(256, 256, 3)$. The entire setup is visualized in Figure 5. These images are stacked at each timestep to form the observations. We excluded the robot states from the inputs, as previous studies have reported that including them can lead to poor performance [7]. The action space is

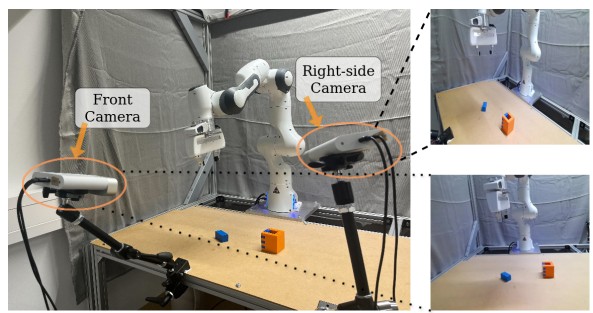

Figure 5: Real robot setup. We use visual inputs from two cameras as observations for policies.

8-dimensional, encompassing the joint positions and the gripper state. The tasks settings detailed below and shown in Figure 6-8. The corresponding results are presented in Table 2-4.

**Pick-Place** We first evaluate our policies using a pick place task. This task contains two individual sub-tasks: place the banana on the plate or place the carrot in the bowl. We collect 30 demonstrations for each individual tasks. During the evaluation, the objects are initialized with similar positions but different orientations.

**Two-stage Pick-Place.** As an enhanced version, this task concatenates the two stages as one manipulation sequence: place both the banana and carrot sequentially to their target areas. Due to the longer motion horizon and more stochasticity, the task complexity significantly increases. The dataset contains 100 demonstrations with objects randomly initialized.

**Inserting.** The robot is required to insert a blue block into a LEGO box. We split this task into two stages: 1) Pick up the block and move it to the box 2) Insert the block inside the box. Since this

task demands precision control which makes it challenging, we fix the LEGO box position and only randomly initialize the state of the block.

**CupStacking.** The robot is required to stack three cups of different sizes. This task contains two stages: 1) Pick up the green cup and place it in the blue cup and 2) Pick up the yellow cup and place it in the green cup. Three cups are aligned almost at the same line at the beginning, then we randomly add variances to each of them. During the evaluation, we use the same initialized positions to evaluate the models and we record the success rate at each stage.

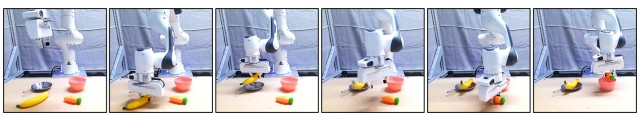

Figure 6: PickPlacing (front camera view)

| DDP-H1 | Pick Place | | Two-stage Pick-Place | |
|--------|--------|--------|---------|---------|
|        | Banana | Carrot | Stage-1 | Stage-2 |
| ED-Tr  | 0.45   | 0.25   | **0.70** | 0.45   |
| ED-Ma  | **0.55** | **0.70** | **0.70** | **0.55** |

Table 2: Pick-Place Success rates

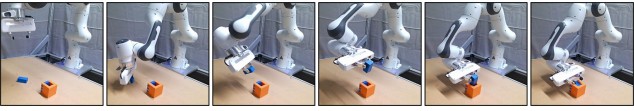

Figure 7: Inserting (right camera view)

| DDP-H1 | Stage-1 | Stage-2 |
|--------|---------|---------|
| ED-Tr  | 0.45    | 0.35    |
| ED-Ma  | **0.65** | **0.6** |

Table 3: Inserting success rates

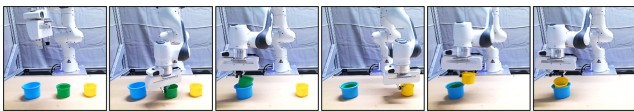

Figure 8: CupStacking (front camera view)

| DDP-H1 | Stack-1 | Stack-2 |
|--------|---------|---------|
| ED-Tr  | 0.6     | 0.4     |
| ED-Ma  | **0.8** | **0.55** |

Table 4: CupStacking success rates

We compared the ED-Tr with ED-Ma using DDP-H1 model. We trained each method for 100 epochs (convergent) and evaluated the model using the final checkpoint. For each task, we performed 20 rollouts with different initial states for the objects. To ensure a fair comparison, we used the same initial states for both Transformer and Mamba evaluations. From the results, Mamba-based methods achieve comparable results with Transformer models.

## 6 Limitations

While MaIL demonstrates excellent performance with smaller dataset sizes, its advantage becomes less pronounced as the dataset scales up. When trained on larger datasets, MaIL achieves results comparable to, but not surpassing, those of Transformer models. Additionally, Mamba is designed to be fast and efficient for large-scale sequences. However, in the context of imitation learning policies where sequences are relatively short, the inference time for Transformers is similar to that of Mamba. This reduces the performance efficiency advantage of Mamba in these scenarios.

## 7 Conclusion

In conclusion, this work presents MaIL, a novel imitation learning (IL) policy architecture that bridges the gap between efficiency and performance in handling sequences of observations. By leveraging the strengths of state space models and rigorously improving upon them, MaIL offers a competitive alternative to the traditionally large and complex Transformer-based policies. The introduction of Mamba in an encoder-decoder structure enhances its versatility, making it suitable for standalone use as well as integration into advanced architectures like diffusion processes. Extensive evaluations on the LIBERO benchmark and real robot experiments demonstrate that MaIL not only matches but also surpasses the performance of existing baselines with limited demonstrations, establishing it as a promising approach for IL tasks.

**Acknowledgments**

This work was supported by funding from the pilot program Core Informatics of the Helmholtz Association (HGF). NS and GN were supported by the Carl Zeiss Foundation under the project JuBot (Jung Bleiben mit Robotern). Xiaogang Jia acknowledges the support from the China Scholarship Council (CSC). The authors acknowledge support by the state of Baden-Württemberg through bwHPC, as well as the HoreKa supercomputer funded by the Ministry of Science, Research and the Arts Baden-Württemberg and by the German Federal Ministry of Education and Research.

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

# A    More Experimental Results

## A.1    Comparison on Different Checkpoints

Evaluating multiple checkpoints can provide important insights into our model's generalization capabilities. Here, we conducted additional experiments assessing the model after 40, 50, and 60 epochs. The results presented in Table 5 suggest that the model neither overfits nor underfits the data at these intervals.

| DDP-H5 | Backbone | w/o language | | | w/ language |
|---|---|---|---|---|---|
| | | LIBERO-Object | LIBERO-Spatial | LIBERO-Long | LIBERO-Goal |
| Epoch 40 | ED-Tr | $0.770_{\pm 0.063}$ | $0.415_{\pm 0.085}$ | $0.345_{\pm 0.044}$ | $0.502_{\pm 0.074}$ |
| | ED-Ma | $0.802_{\pm 0.030}$ | $0.478_{\pm 0.039}$ | $0.355_{\pm 0.026}$ | $\mathbf{0.615}_{\pm \mathbf{0.028}}$ |
| Epoch 50 | ED-Tr | $0.758_{\pm 0.045}$ | $0.412_{\pm 0.060}$ | $0.333_{\pm 0.023}$ | $0.497_{\pm 0.072}$ |
| | ED-Ma | $\mathbf{0.807}_{\pm \mathbf{0.051}}$ | $\mathbf{0.500}_{\pm \mathbf{0.031}}$ | $0.385_{\pm 0.023}$ | $0.580_{\pm 0.041}$ |
| Epoch 60 | ED-Tr | $0.767_{\pm 0.045}$ | $0.448_{\pm 0.092}$ | $0.348_{\pm 0.050}$ | $0.505_{\pm 0.036}$ |
| | ED-Ma | $0.805_{\pm 0.058}$ | $0.497_{\pm 0.050}$ | $\mathbf{0.390}_{\pm \mathbf{0.028}}$ | $0.610_{\pm 0.035}$ |
| Max | ED-Tr | $0.770_{\pm 0.063}$ | $0.448_{\pm 0.092}$ | $0.348_{\pm 0.05}$ | $0.505_{\pm 0.036}$ |
| | ED-Ma | $\mathbf{0.807}_{\pm \mathbf{0.051}}$ | $\mathbf{0.500}_{\pm \mathbf{0.031}}$ | $\mathbf{0.390}_{\pm \mathbf{0.028}}$ | $\mathbf{0.615}_{\pm \mathbf{0.028}}$ |

Table 5: Comparison on different checkpoints based on DDP-H5 methods.

## A.2    Comparison on Full Demonstrations

To investigate how MaIL and Transformers scale with dataset size, we evaluated both models on the LIBERO benchmark with access to the full dataset. Due to time constraints, we focused solely on the Encoder-Decoder architecture for both MaIL and Transformers. We conducted experiments across different checkpoints, evaluating each model at three checkpoints and reporting their best performance. The results are presented in Table 6:

| Policy | Backbone | w/o language | | | w/ language |
|---|---|---|---|---|---|
| | | LIBERO-Object | LIBERO-Spatial | LIBERO-Long | LIBERO-Goal |
| BC-H1 | ED-Tr | $0.847_{\pm 0.020}$ | $0.658_{\pm 0.051}$ | $\mathbf{0.658}_{\pm \mathbf{0.035}}$ | $0.668_{\pm 0.055}$ |
| | ED-Ma | $\mathbf{0.862}_{\pm \mathbf{0.036}}$ | $\mathbf{0.693}_{\pm \mathbf{0.061}}$ | $0.642_{\pm 0.011}$ | $\mathbf{0.793}_{\pm \mathbf{0.050}}$ |
| BC-H5 | ED-Tr | $\mathbf{0.910}_{\pm \mathbf{0.018}}$ | $\mathbf{0.733}_{\pm \mathbf{0.049}}$ | $0.632_{\pm 0.056}$ | $0.772_{\pm 0.026}$ |
| | ED-Ma | $0.907_{\pm 0.019}$ | $0.697_{\pm 0.062}$ | $\mathbf{0.687}_{\pm \mathbf{0.038}}$ | $\mathbf{0.833}_{\pm \mathbf{0.015}}$ |
| DDP-H1 | ED-Tr | $\mathbf{0.918}_{\pm \mathbf{0.028}}$ | $0.707_{\pm 0.035}$ | $0.683_{\pm 0.015}$ | $0.788_{\pm 0.056}$ |
| | ED-Ma | $0.913_{\pm 0.034}$ | $\mathbf{0.725}_{\pm \mathbf{0.035}}$ | $\mathbf{0.700}_{\pm \mathbf{0.015}}$ | $\mathbf{0.818}_{\pm \mathbf{0.016}}$ |
| DDP-H5 | ED-Tr | $\mathbf{0.952}_{\pm \mathbf{0.003}}$ | $0.745_{\pm 0.018}$ | $0.731_{\pm 0.031}$ | $0.873_{\pm 0.030}$ |
| | ED-Ma | $0.932_{\pm 0.029}$ | $\mathbf{0.758}_{\pm \mathbf{0.045}}$ | $\mathbf{0.772}_{\pm \mathbf{0.004}}$ | $\mathbf{0.893}_{\pm \mathbf{0.034}}$ |

Table 6: Performance on LIBERO benchmark with access to the full dataset.

## A.3    Experiments with Language Instructions

In Table 1 of our manuscript, we did not utilize language instructions for the LIBERO-Object, LIBERO-Spatial, and LIBERO-Long tasks. To gain further insights, we conducted additional experiments on these tasks using language embeddings, with access to 20% and 100% of the data. The results are presented in Tables 7 and 8:

|  |  | w/ language | | |
| Policy | Backbone | LIBERO-Object | LIBERO-Spatial | LIBERO-Long |
| BC-H1 | ED-Tr | $0.596_{\pm0.123}$ | $0.474_{\pm0.117}$ | $0.316_{\pm0.035}$ |
|  | ED-Ma | $\mathbf{0.689}_{\pm\mathbf{0.062}}$ | $\mathbf{0.518}_{\pm\mathbf{0.080}}$ | $\mathbf{0.408}_{\pm\mathbf{0.050}}$ |
| BC-H5 | ED-Tr | $0.714_{\pm0.036}$ | $\mathbf{0.426}_{\pm\mathbf{0.035}}$ | $0.278_{\pm0.066}$ |
|  | ED-Ma | $\mathbf{0.764}_{\pm\mathbf{0.075}}$ | $0.395_{\pm0.083}$ | $\mathbf{0.303}_{\pm\mathbf{0.115}}$ |
| DDP-H1 | ED-Tr | $0.761_{\pm0.005}$ | $0.521_{\pm0.002}$ | $0.396_{\pm0.031}$ |
|  | ED-Ma | $\mathbf{0.781}_{\pm\mathbf{0.045}}$ | $\mathbf{0.576}_{\pm\mathbf{0.022}}$ | $\mathbf{0.499}_{\pm\mathbf{0.048}}$ |
| DDP-H5 | ED-Tr | $\mathbf{0.864}_{\pm\mathbf{0.025}}$ | $\mathbf{0.618}_{\pm\mathbf{0.055}}$ | $0.431_{\pm0.010}$ |
|  | ED-Ma | $0.854_{\pm0.040}$ | $0.609_{\pm0.050}$ | $\mathbf{0.460}_{\pm\mathbf{0.035}}$ |

Table 7: Performance on LIBERO benchmark with 20% data.

|  |  | w/ language | | |
| Policy | Backbone | LIBERO-Object | LIBERO-Spatial | LIBERO-Long |
| BC-H1 | ED-Tr | $0.803_{\pm0.043}$ | $0.718_{\pm0.020}$ | $0.724_{\pm0.072}$ |
|  | ED-Ma | $\mathbf{0.859}_{\pm\mathbf{0.027}}$ | $\mathbf{0.776}_{\pm\mathbf{0.075}}$ | $\mathbf{0.770}_{\pm\mathbf{0.054}}$ |
| BC-H5 | ED-Tr | $0.916_{\pm0.027}$ | $0.800_{\pm0.018}$ | $\mathbf{0.764}_{\pm\mathbf{0.039}}$ |
|  | ED-Ma | $\mathbf{0.929}_{\pm\mathbf{0.017}}$ | $\mathbf{0.838}_{\pm\mathbf{0.048}}$ | $0.758_{\pm0.070}$ |
| DDP-H1 | ED-Tr | $\mathbf{0.916}_{\pm\mathbf{0.025}}$ | $\mathbf{0.813}_{\pm\mathbf{0.082}}$ | $0.763_{\pm0.037}$ |
|  | ED-Ma | $0.901_{\pm0.020}$ | $0.743_{\pm0.043}$ | $\mathbf{0.786}_{\pm\mathbf{0.030}}$ |
| DDP-H5 | ED-Tr | $\mathbf{0.955}_{\pm\mathbf{0.005}}$ | $0.836_{\pm0.045}$ | $0.826_{\pm0.035}$ |
|  | ED-Ma | $0.943_{\pm0.050}$ | $\mathbf{0.883}_{\pm\mathbf{0.005}}$ | $\mathbf{0.830}_{\pm\mathbf{0.063}}$ |

Table 8: Performance on LIBERO benchmark with full data.

# B   Additional Task Details

## B.1   LIBERO benchmark

- *LIBERO-Spatial* contains various objects and different spatial relationships among those objects within a consistent layout. This suite challenges the robot's spatial understanding and its ability to navigate and manipulate objects based on their spatial configurations.

- *LIBERO-Object* suite has tasks involving different objects on the same layout, requiring the robot to pick-place a unique object at a time. It emphasizes the policy's ability to accurately recognize various types of objects.

- *LIBERO-Goal* suite maintains the same objects and fixed spatial relationships but varies the task goals. This suite challenges the robot's ability to understand and achieve different objectives despite the consistent arrangement of objects.

- *LIBERO-Long* suite focuses on long-horizon tasks, evaluating the robot's ability to perform and manage extended sequences of actions over a prolonged period.

- *LIBERO-90* contains 90 short-horizon tasks with significant diversity. This suite includes a wide variety of object categories, layouts, and goals, providing a comprehensive evaluation of the robot's adaptability and versatility in handling diverse scenarios.

## B.2   Real Robot Tasks

### B.2.1   CupStacking

In the CupStacking task, there are three cups with differing sizes which the robot needs to stack inside one another. The large cup is on the left, the small cup is on the right and the middle-sized cup is in the middle. In the training data, only one of the two modes is covered: the middle-sized cup is put into the large cup first, and the small cup is stacked afterwards. The horizontal positions of the cups are fixed, but their vertical positions can vary.

### B.2.2 Inserting

In the Inserting task, there is a LEGO box with a hole in the middle and another block which can fit in the hole of the LEGO box. The goal in this task is to grab the block and insert it in the hole of the LEGO box. The position and orientation of the LEGO box are fixed, but the position and orientation of the block can vary inside a specific area on the table.

### B.2.3 PickPlacing

In the PickPlacing task, there are two different bowls, a banana and a carrot. The goal is to pick the banana and the carrot and place them inside the bowls. The banana is always placed in the left bowl, and the carrot is placed in the right bowl. The positions and orientations of the bowls are fixed, but the positions and orientations of the banana and the carrot can vary inside a specific area on the table.

## C  Architectures

### C.1  State Space Models

Inspired by the Attention mechanism [15] in Transformers, Mamba [13] improves upon the Structured State-Space Sequence (S4) Model by using a selective scan operator, to propagate or forget information over context, allowing it to filter out relevant features. Specifically, the S4 model is inspired by a continuous system, $x(l) \in \mathbb{R}^{\mathrm{D}} \mapsto y(l) \in \mathbb{R}^{\mathrm{D}}$, that maps $x(l)$ into a hidden state $h(l) \in \mathbb{R}^{\mathrm{N}}$, which then can be projected onto the output $y(t)$, that is,

$$y(l) = \mathbf{C}h(l) \quad \text{with} \quad h'(l) = \mathbf{A}h(l) + \mathbf{B}x(l), \tag{6}$$

where $\mathbf{A} \in \mathbb{R}^{\mathrm{N} \times \mathrm{N}}$ are the evolution parameters, $\mathbf{B} \in \mathbb{R}^{\mathrm{N} \times \mathrm{D}}$ and $\mathbf{C} \in \mathbb{R}^{\mathrm{D} \times \mathrm{N}}$ are the projection parameters. S4 transforms the parameters $(\mathbf{A}, \mathbf{B})$ to discrete parameters $(\overline{\mathbf{A}}, \overline{\mathbf{B}})$ via

$$\overline{\mathbf{A}} = \exp\left(\mathbf{\Delta A}\right) \quad \text{and} \quad \overline{\mathbf{B}} = (\mathbf{\Delta A})^{-1}(\exp\left(\mathbf{\Delta A}\right) - \mathbf{I}) \cdot \mathbf{\Delta B}, \tag{7}$$

where $\mathbf{\Delta}$ is the step size. The model then can be computed as either linear recurrence, that is,

$$y_t = \mathbf{C}h_t, \quad \text{with} \quad h_t = \overline{\mathbf{A}}h_{t-1} + \overline{\mathbf{B}}x_t, \tag{8}$$

or through a global convolution $*$ given by

$$\mathbf{y} = \mathbf{x} * \overline{\mathbf{K}}, \quad \text{with} \quad \overline{\mathbf{K}} = (\mathbf{C}\overline{\mathbf{B}}, \mathbf{C}\overline{\mathbf{A}}\overline{\mathbf{B}}, \dots, \mathbf{C}\overline{\mathbf{A}}^k\overline{\mathbf{B}}). \tag{9}$$

### C.2  Transformers

Here we depict two transformer-based architectures in diffusion policy: a decoder-only model (Figure 9) and an encoder-decoder model (Figure 10). Both architectures leverage the strengths of transformer models to effectively handle sequential data and capture long-range dependencies.

## D  Model Details

### D.1  Parameter Comparison

For all Mamba-based policies, we fix the expansion factor at 2. The parameters exhibit slight variance when using the same number of layers due to differences in the choice of hidden state values and convolution widths. Therefore, at the same layer depth, we use the average parameter values to account for these variations. We list the comparison of parameters for all models in Table 10.

We also evaluate the inference time on a local PC equipped with an RTX 2060 GPU, using a batch size of 32 to ensure all models are assessed under the same conditions in Table 9.

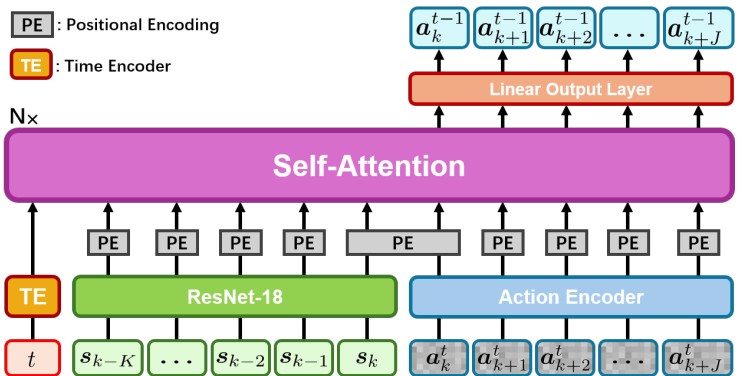

Figure 9: Decoder-only learning block. This architecture integrates ResNet-18 for state encoding and an action encoder for actions with horizon $J$, with both components feeding into a self-attention mechanism. Positional encoding (PE) and time encoding (TE) enhance the inputs. Ultimately, the output of self-attention is fed to a linear output layer to predict future actions.

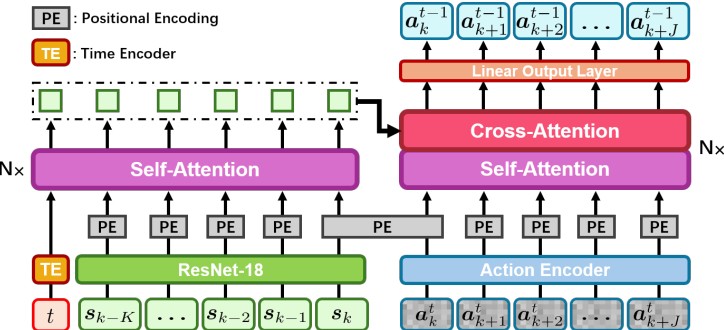

Figure 10: Encoder-decoder learning block. This figure illustrates the architecture of an encoder-decoder transformer block designed for policy learning. In the encoder, states are encoded using ResNet-18, enhanced by time encoding (TE) and positional encoding (PE), and processed through self-attention. The decoder then utilizes self-attention on the encoded actions and employs cross-attention to integrate the encoded states from the encoder. Ultimately, the output of cross-attention is fed to a linear output layer to predict future actions.

## D.2   Training Details

We list the training hyperparameters for the Transformer-based and Mamba-based policies in Table 10. To ensure a fair comparison, we tune the hyperparameters of both policies at the same level.

The policies are trained using the human expert demonstrations provided by LIBERO, where we only take 10 demonstrations for each task in main experiment.

All models are trained on a cluster equipped with 4 A100 GPUs, with a batch size of 256, over 50 epochs using 3 different seeds. Finally, we calculate the average success rate across these 3 seeds.

## D.3   Ablation on Observation Occlusions

For the image-based model, recent observations are not significantly different from the current observation. To evaluate the influence of historical data and compel the model to focus more on past observations, we employ occlusion techniques on the observations. This involves randomly masking portions of the images like the Figure 11. The occlusion rate determines the extent to which the images are masked.

| Method | Layers | Params | Inference time (H1) | Inference time (H5) |
|--------|--------|--------|---------------------|---------------------|
| BC     |        |        |                     |                     |
| D-Tr   | 6      | 23.6M  | 0.03043             | 0.05906             |
|        | 8      | 24.0M  | 0.03305             | 0.06306             |
| ED-Tr  | 4 + 4  | 24.2M  | 0.03230             | 0.06368             |
|        | 4 + 6  | 24.7M  | 0.03505             | 0.06563             |
| D-Ma   | 10     | 23.6M  | 0.02603             | 0.05340             |
|        | 12     | 23.8M  | 0.02797             | 0.05477             |
|        | 16     | 24.3M  | 0.02855             | 0.05614             |
| ED-Ma  | 6 + 6  | 23.8M  | 0.02752             | 0.05465             |
|        | 6 + 8  | 24.0M  | 0.02833             | 0.05733             |
|        | 6 + 10 | 24.2M  | 0.02974             | 0.05862             |
|        | 8 + 10 | 24.4M  | 0.03074             | 0.06007             |
|        | 10 + 10| 24.6M  | 0.03190             | 0.06112             |
| DDP    |        |        |                     |                     |
| D-Tr   | 6      | 23.7M  | 0.10300             | 0.14074             |
|        | 8      | 24.1M  | 0.12882             | 0.16428             |
| ED-Tr  | 4 + 4  | 24.3M  | 0.14443             | 0.17995             |
|        | 4 + 6  | 24.8M  | 0.17244             | 0.21022             |
| D-Ma   | 10     | 23.7M  | 0.11185             | 0.14384             |
|        | 12     | 23.9M  | 0.12616             | 0.16145             |
|        | 16     | 24.2M  | 0.15699             | 0.18236             |
| ED-Ma  | 6 + 6  | 23.8M  | 0.12653             | 0.15984             |
|        | 6 + 8  | 24.0M  | 0.14025             | 0.17350             |
|        | 6 + 10 | 24.3M  | 0.15977             | 0.19398             |
|        | 8 + 10 | 24.5M  | 0.16883             | 0.20553             |
|        | 10 + 10| 24.7M  | 0.18332             | 0.22571             |

Table 9: Inference Time Comparison, H1 and H5 refer to using current state and 5 steps historical states respectively.

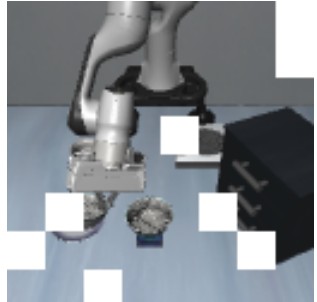 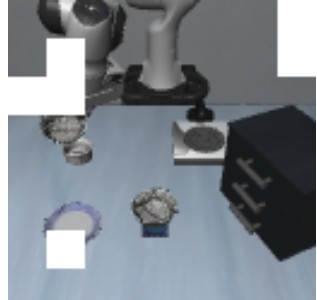 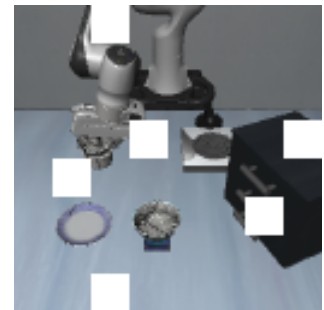

Figure 11: Visualization of Observation Occlusions

| Methods / Parameters | libero_object | libero_spatial | libero_long | libero_goal | libero_90 |
|---|---|---|---|---|---|
| **history = 5** | | | | | |
| **D-Ma** | | | | | |
| Number of Mamba Layer | 10 | 16 | 16 | 12 | 16 |
| Hidden State Dimension | 16 | 8 | 16 | 8 | 16 |
| Convolution Width | 4 | 4 | 4 | 2 | 4 |
| Expand Factor | 2 | 2 | 2 | 2 | 2 |
| **ED-Ma** | | | | | |
| Number of Encoder Layer | 8 | 10 | 8 | 10 | 8 |
| Number of Decoder Layer | 10 | 10 | 10 | 10 | 10 |
| Hidden State of Encoder | 8 | 8 | 8 | 8 | 8 |
| Hidden State of Decoder | 8 | 16 | 8 | 16 | 8 |
| Encoder Convolution Width | 4 | 4 | 2 | 2 | 2 |
| Decoder Convolution Width | 2 | 2 | 2 | 2 | 2 |
| Expand Factor | 2 | 2 | 2 | 2 | 2 |
| **D-Tr** | | | | | |
| Number of Attention Block | 8 | 8 | 8 | 8 | 8 |
| Number of Attention Head | 4 | 4 | 4 | 4 | 4 |
| **ED-Tr** | | | | | |
| Number of Encoder Block | 4 | 4 | 4 | 4 | 4 |
| Number of Decoder Block | 6 | 6 | 6 | 6 | 6 |
| Number of Encoder Head | 4 | 4 | 4 | 4 | 4 |
| Number of Decoder Head | 4 | 4 | 4 | 4 | 4 |
| **history = 1** | | | | | |
| **D-Ma** | | | | | |
| Number of Mamba Layer | 10 | 16 | 16 | 12 | 16 |
| Hidden State Dimension | 16 | 8 | 16 | 8 | 16 |
| Convolution Width | 4 | 4 | 4 | 2 | 4 |
| Expand Factor | 2 | 2 | 2 | 2 | 2 |
| **ED-Ma** | | | | | |
| Number of Encoder Layer | 10 | 10 | 10 | 10 | 8 |
| Number of Decoder Layer | 8 | 10 | 10 | 8 | 10 |
| Hidden State of Encoder | 16 | 16 | 8 | 16 | 8 |
| Hidden State of Decoder | 8 | 8 | 16 | 16 | 8 |
| Encoder Convolution Width | 2 | 4 | 2 | 2 | 2 |
| Decoder Convolution Width | 4 | 2 | 4 | 4 | 2 |
| Expand Factor | 2 | 2 | 2 | 2 | 2 |
| **D-Tr** | | | | | |
| Number of Attention Block | 8 | 8 | 8 | 8 | 8 |
| Number of Attention Head | 4 | 4 | 4 | 4 | 4 |
| **ED-Tr** | | | | | |
| Number of Encoder Block | 4 | 4 | 4 | 4 | 4 |
| Number of Decoder Block | 6 | 6 | 6 | 6 | 6 |
| Number of Encoder Head | 4 | 4 | 4 | 4 | 4 |
| Number of Decoder Head | 4 | 4 | 4 | 4 | 4 |

Table 10: Hyperparameter in Simulated Experiments

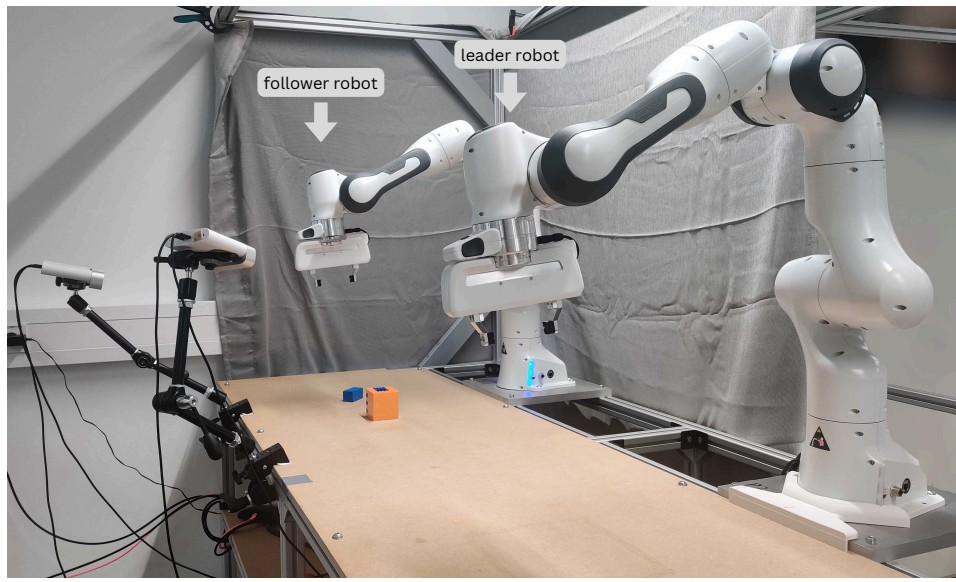

Figure 12: Data collection scene for real robot

# E Real Robot Details

## E.1 Setup

The current setup consists of two robots and two computers. One of the computers, the robot PC, runs a real-time OS to control the robots reliably using Polymetis (E.2), which is a real-time Pytorch controller manager for robots. The robot PC uses a PD controller with the desired joint position as a configurable parameter. The robot PC hosts a server from which the desired joint position and the gripper of the robot can be configured. The other PC, the workstation, runs the models and sends the desired joint position and gripper state to the robot PC at each time step. The workstation also has access to two cameras, which are used to capture images as input for the models.

## E.2 Polymetis

Polymetis [45] is a software framework designed to improve the control and interaction of robotic systems. It connects high-level decision-making with low-level hardware control, allowing robots to perform complex tasks like assembly and inspection accurately. Key features include real-time motion planning, adaptive control, and sensory feedback. Polymetis supports various robotic platforms and sensors, making it useful for both research and industry. Its simplicity, detailed documentation, and strong API make Polymetis a valuable tool for enhancing robotic capabilities and intelligence.

## E.3 Data Collection

Teleoperation is used to collect data for all the real robot tasks, where the leader robot is controlled by a human and the follower robot follows the leader robot as shown in Figure 12. The objects are put in front of the follower robot and the cameras do not see the leader robot or the human. The current joint state of the leader robot is sent to the follower robot as the desired joint state. The state of the gripper is considered to be binary, either closed or open. A threshold is set for the gripper of the leader robot; if the current width is below the threshold, the gripper of the follower robot closes, otherwise it opens.

### E.4 Evaluation

For the evaluation, using the output of the model sometimes activate the security mechanism of the robot because it violates a certain constraint. To solve this issue, a trajectory is generated between the current joint position and the output of the model. The points of this trajectory are then given to the robot at each time step instead of the raw output of the model. The length of this trajectory varies depending on how far away the output of the model is from the current robot state.

## F  Analysis on the State Space Representations

To better understand the advantages of MaIL over Transformers, we conducted a detailed analysis of the latent representations produced by both methods. Specifically, we compared the Encoder-Decoder architectures of MaIL and Transformers by visualizing their high-dimensional representations using t-SNE [46]. We employed BC-based models trained on the LIBERO-Object dataset using full demonstrations and rolled out the models across entire trajectories. The visualizations represent the latent spaces before the final action prediction layer (linear). Figure 13 displays the t-SNE results for four different trajectory representations.

As shown in the figure, MaIL's state representations are notably more structured across all trajectories compared to those generated by Transformers. More structured representations could potentially improve the generalization of MaIL, which explains the advantages of MaIL over Transformers.

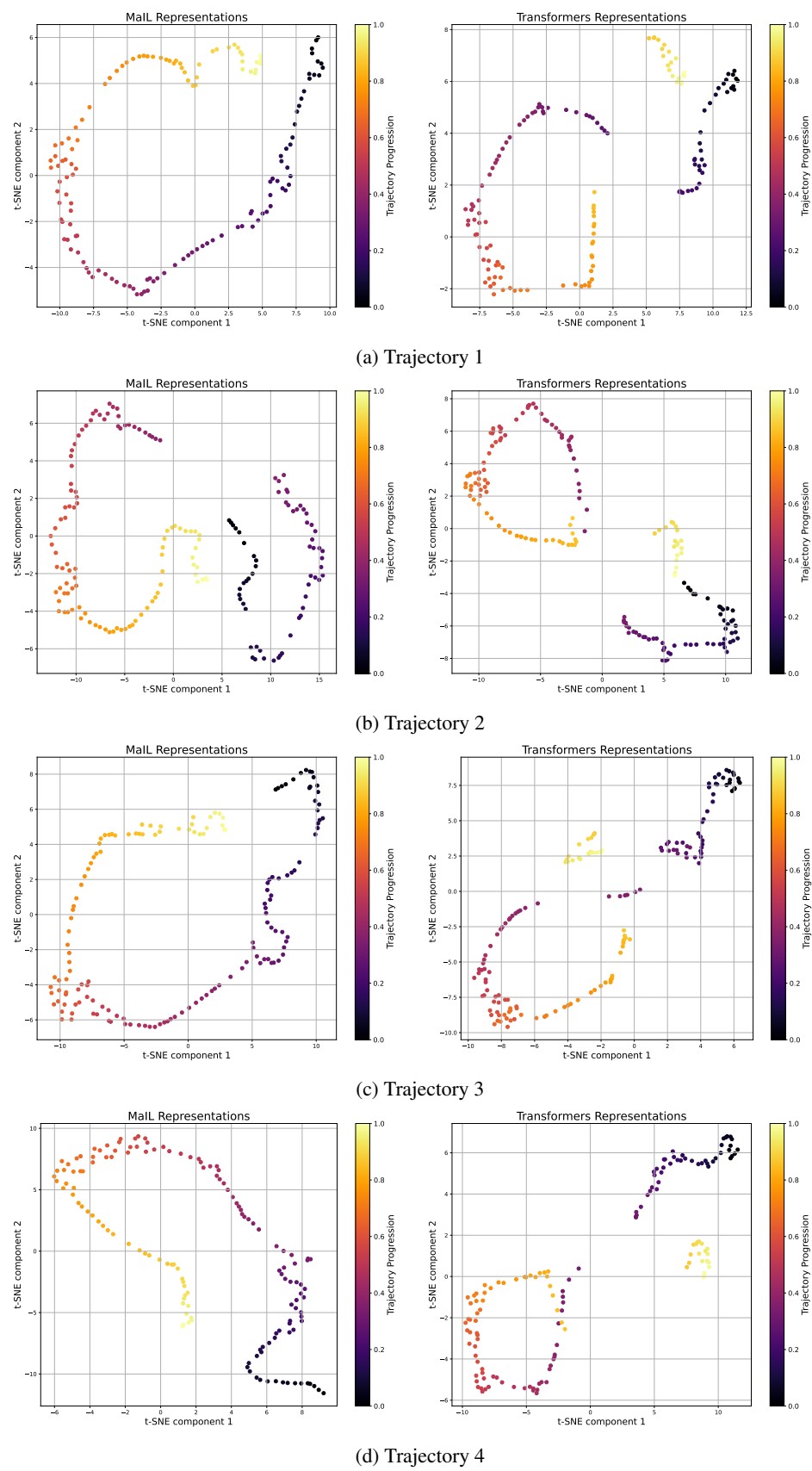

(a) Trajectory 1

(b) Trajectory 2

(c) Trajectory 3

(d) Trajectory 4

Figure 13: Visualization of latent representations on different trajectories.

