# OpenReview forum: "MaIL: Improving Imitation Learning with Selective State Space Models"
_robot-learning.org/CoRL/2024/Conference — CoRL 2024_

### Official Review · Reviewer_Vk9K · 2024-07-19
**Good paper**

**Originality:** 3
**Technical Quality:** 3
**Clarity Of Presentation:** 4
**Potential Impact:** 3
**Recommendation:** 3
**Confidence:** 3

**Review:**

My opinion is positive. The paper explores the utilization of Mamba, a novel state space model which is an alternative to Transformers. This work proposes using Mamba in imitation learning with selective state space models (MaIL). They propose two possible architectures of MaIL: an encoder-only Mamba denoising architecture (D-Ma) and a Mamba encoder-decoder architecture (ED-Ma).

In addition:
- The manuscript is well-organized and written, and the goals and contributions of the authors are clearly stated.
- The research questions they want to answer with their experiments are clearly stated and discussed.
- The figures and algorithms used in the paper complement the text and help to understand the authors' explanations.
- The appendix section includes valuable information regarding the transformer architecture, ablation studies, model, and experimental details.
- The authors clearly explain the limitations of their method when the dataset scales up and the advantage compared to Transformers is less noticeable.

However, I recommend that the authors expand the discussion of the experimental results, especially the cases in which D-Ma is outperformed by ED-Ma in LIBERO-Long and the possible reasons for this.
Also, the authors did not sufficiently evaluate Mamba's ability to handle long-term contextual information. They only tested H5 conditions.
Considering that Mamba is an SSM, the obtained state space representations are better analyzed qualitatively.

**Quality Of The Limitations Section:**

3

**Questions For Rebuttal:**

- How are the performances of D-Ma and ED-Ma affected by the historical states? In the experiments, ED-Ma outperforms the others in BC-H1; however, when historical data is added in BC-H5, D-Ma outperforms ED-Ma.
- Why is the performance of ED-Ma lower than ED-Tr in stage 2 of the Two-stage Pick-Place and Inserting tasks?
- In the ablation studies of BC-H1 and BC-H5, at 60% and 100%, the success rates of ED-Ma and D-Ma were very similar to those of D-Tr and ED-Tr. What could be the reason for this?
- Why, in the ablation studies BC-H5: spatial at 100%, is the success rate of D-Ma lower than the other methods? Its success rate at 100% seems to be the same as at 60%.
- In Figure 4 (b), the label is DDP-H5. Is this correct? Should it not be DDP-H5: object?
- Could you have additional discussion over the application of the proposed method to BC with longer context?
- Could you analyze the representations of the formed state space qualitatively?

**Robotics Focus:**

3

**Summary Of Paper:**

This paper introduces MaIL (Mamba Imitation Learning), a novel approach that utilizes Mamba, a computationally efficient alternative to transformer-based models, for imitation learning. MaIL is evaluated against Behavior Cloning and Denoising Diffusion Policies, demonstrating superior performance. Extensive experiments in both simulation and real-world environments confirm that MaIL not only matches but often surpasses existing baseline methods.

**Summary Of Recommendation:**

My opinion is positive, and I would accept this manuscript for publication as it explores an exciting and novel approach of using Mamba in Imitation Learning. However, I suggest the authors to improve the discussion of the results of their experiments.

---

### Official Review · Reviewer_mQZR · 2024-07-20
**Novel architecture but results not convincing**

**Originality:** 4
**Technical Quality:** 4
**Clarity Of Presentation:** 4
**Potential Impact:** 3
**Recommendation:** 3
**Confidence:** 5

**Review:**

Strengths:
The paper is well-written with the contributions explained clearly throughout the paper. The usage of this architecture is novel and can potentially challenge the state-of-the-art Transformer backbones for visuomotor imitation learning.

Weaknesses:
1. I am a bit worried about the quantitative results in this paper. Success rates shown in Table 1 on the LIBERO multi-task benchmark are mostly and roughly in the range 0.3 to 0.5, which I am not convinced is a good success rate on this benchmark, and seemingly at par or even worse than that in the LIBERO’s paper itself. MaIL and the considered transformer architecture’s success rates are especially low on the LIBERO-Long suite of tasks, which does not seem to convey that MaIL seems to solve the problem with processing long sequences of observations efficiently.
2. The authors mentioned in Section 5.2 that all models were trained for 50 epochs - can the authors clarify what they considered as an epoch, i.e. did it comprise of all possible sequences that can be chunked out of the dataset of demos and if there was any overlap between obs-action sequences the model was trained on. I think it would also be best for reproducibility if the authors mentioned the number of gradient steps the models were trained on.
3. Given that the authors present results having evaluated just one checkpoint, I would like to understand if the model fit/overfit/underfit the data at that checkpoint - something that I believe could be overlooked if multiple checkpoints were evaluated.
4. Why was BC-RNN not considered as a baseline?

**Quality Of The Limitations Section:**

3

**Questions For Rebuttal:**

Please refer to the weaknesses mentioned above.

**Robotics Focus:**

4

**Summary Of Paper:**

The paper presents a new architecture for handling sequences of observations in imitation learning, leveraging Mamba as the backbone. In their experiments, the authors try to assess this architecture’s performance for visuomotor imitation learning compared to a Transformer backbone, and the ability to handle multi-modal instructions such as language instructions. The authors present results on the LIBERO multi-task benchmark in simulation as well as on a real robot.

**Summary Of Recommendation:**

I believe the paper takes a stride into an important direction of discovering a better backbone for visuomotor imitation learning. However, I recommend a weak reject for this paper as I am not convinced that the quantitative results in simulation suggest that this architecture is sufficiently superior than the Transformer backbone. As I mentioned in my review, I believe the evaluation protocol is suboptimal and takeaways might change had evaluation been done differently. Additionally, there are also some missing baselines.

---

### Official Review · Reviewer_u9Yp · 2024-07-22
**Improving Imitation Learning via Mamba**

**Originality:** 3
**Technical Quality:** 3
**Clarity Of Presentation:** 3
**Potential Impact:** 3
**Recommendation:** 3
**Confidence:** 3

**Review:**

**Strengths**

* The idea and motivation of the paper are solid. Mamba has recently received significant attention for matching Transformers' performance on some benchmarks with lower memory usage and better speed. It makes sense to study the performance of Mamba-based imitation policies in robotics.

* The authors have proposed both decoder-only and encoder-decoder architectures, similar to Transformers, and evaluated them in both behavior cloning and denoising diffusion policies. The authors propose a novel approach called Mamba Aggregation for the encoder-decoder variant.

* The proposed approach outperforms Transformer-based counterparts on 5 LIBERO suites, some including language modality, when having access to a small portion of datasets (20%).

* They have conducted real-world experiments on three different tasks, and in all of them, MaIL outperforms the baseline.


**Weaknesses**

* Although the paper is well-written and easy to follow, it can be misleading at times. In the abstract, the caption of Table 1, and the conclusion, the authors claim MaIL outperforms Transformers in all LIBERO tasks without clarifying that the evaluations are conducted with all models having access to only 20% of the dataset.

* The ablation on dataset size is not sufficiently informative. The authors compare their approach with the baseline using the full dataset on only two tasks from the five task suites and only with the BC variants. A comprehensive comparison table, similar to Table 1, using the full dataset across all tasks, would have been more insightful.

* Although one of the main contributions of the paper is the Mamba Aggregation approach in the Encoder-Decoder architecture, the authors do not elaborate much on their inspirations and design choices for this.

**Quality Of The Limitations Section:**

3

**Questions For Rebuttal:**

* How do Transformer-based baselines compare with your approach when having access to the full LIBERO dataset?
* Could you please elaborate more on the Aggregation approach? How are the state, action, and time variables learned? What are the advantages and disadvantages of this approach?

**Robotics Focus:**

4

**Summary Of Paper:**

This paper presents Mamba Imitation Learning (MaIL) as an efficient alternative to Transformer-based policies, leveraging the Mamba state space model.  MaIL leverages Mamba's advantages and matches Transformer performance with better speed, memory usage, and efficiency.  The authors offer two variants: - Decoder-only: Denoises actions using noised inputs and time embeddings. - Encoder-decoder: Handles multi-modal inputs with learnable embeddings.  MaIL outperforms Transformers on LIBERO benchmarks with 20% of the dataset, processes multi-modal inputs, and is robust to noise.  The authors demonstrate the effectiveness of MaIL in three real robot tasks.

**Summary Of Recommendation:**

Given that this paper explores the benefits of the recent Mamba architecture in the context of imitation learning, both in simulation and real-world setups, I find it to be a valuable contribution to the field. Therefore, I recommend accepting this paper.

---

### Official Review · Reviewer_k2cG · 2024-07-23
**Replacing Transformers in IL with Mamba**

**Originality:** 2
**Technical Quality:** 2
**Clarity Of Presentation:** 3
**Potential Impact:** 3
**Recommendation:** 2
**Confidence:** 3

**Review:**

## Overview

The paper introduces the Mamba for Imitation Learning (MaIL), presenting a novel approach by incorporating the Mamba model with established IL techniques such as Behavior Cloning and Denoising Diffusion Policies. This integration is explored through two architectural variants: a decoder-only model and an encoder-decoder model. The study provides a comprehensive evaluation of these models through both simulation and real-world experiments. Results demonstrate that the Mamba architecture for Imitation Learning (MaIL) significantly outperforms traditional Transformer-based methods in IL, showcasing its effectiveness and adaptability in both simulated and practical settings.

Overall, this paper presents a timely "A+B" style contribution but lacks a clear explanation of the underlying mechanisms that drive its success. A more detailed exploration into why MaIL works would greatly enhance the depth and impact of the findings.


## Strengths

- The utilization of the Mamba model within the realm of imitation learning, while intuitive, proposes a compelling alternative to the prevailing Transformer-based methodologies.

- Extensive evaluations are conducted utilizing both simulated benchmarks (LIBERO) and real robotic tasks, effectively demonstrating the method's adaptability and efficiency across diverse scenarios.

- MaIL consistently surpasses Transformer-based approaches in various tasks, notably under conditions of limited data availability, a critical factor for practical robotic applications.

- Demonstrating proficiency in managing multi-modal inputs, such as images and language, MaIL proves to be highly effective for a wide range of complex and versatile robotic tasks.

## Weaknesses

- While the empirical evidence underscores MaIL's robust performance against Transformer-based architectures, the underlying reasons of such advantages remain ambiguous. A deeper exploration into why Mamba excels would enhance the understanding of its efficacy.

- A notable strength of Transformer architectures is their ability to manage long-horizon dependencies. The capability of MaIL to handle similar tasks remains uncertain. Testing MaIL on tasks that require extensive long-horizon reasoning could verify its potential in more complex scenarios.

- The Mamba architecture is noted for its speed advantage, particularly when dealing with very long sequences, as per the original Mamba paper. However, in typical imitation learning scenarios where sequence lengths are much shorter, the relevance of this speed benefit is questionable. Clarifying whether Mamba retains its efficiency advantage in standard imitation learning contexts would be valuable.

- MaIL's effectiveness seems to converge with that of Transformers when large datasets are employed, which may diminish its advantages in data-abundant situations. Given that Transformers excel at scaling with large volumes of data [1,2,3], further investigation into MaIL's performance with substantial datasets is warranted.

- The claim that "It is evident that Mamba-based models significantly outperform Transformers when data is scarce and perform comparably as the dataset size increases" is intriguing yet lacks detailed analysis or explanation. Providing a thorough examination or empirical backing for this observation would substantiate the claim and clarify the conditions under which MaIL is most effective.

[1] Brohan, Anthony, et al. "Rt-1: Robotics transformer for real-world control at scale." _arXiv preprint arXiv:2212.06817_ (2022).

[2] Brohan, Anthony, et al. "Rt-2: Vision-language-action models transfer web knowledge to robotic control." _arXiv preprint arXiv:2307.15818_ (2023).

[3] Kim, Moo Jin, et al. "OpenVLA: An Open-Source Vision-Language-Action Model." _arXiv preprint arXiv:2406.09246_ (2024).

**Quality Of The Limitations Section:**

3

**Questions For Rebuttal:**

- Why do the authors emphasize MaIL's effective handling of multi-modal sensory input? I would assume this capability also extends to Transformer models, correct?

**Robotics Focus:**

4

**Summary Of Paper:**

**Main Idea:** Employ the Mamba architecture for Imitation Learning (IL).  **Main Contributions:** - Integrates the Mamba model with Behavior Cloning and Denoising Diffusion Policies. - Furthermore, the study introduces two architectural variants: decoder-only and encoder-decoder. - Both simulation and real-world experiments validate the stronger performance of MaIL over Transformer-based approaches in IL.

**Summary Of Recommendation:**

Overall, this paper presents a timely "A+B" style contribution but lacks a clear explanation of the underlying mechanisms that drive its success. A more detailed exploration into why MaIL works would greatly enhance the depth and impact of the findings.

---

### Author Rebuttal · Authors · 2024-08-13

We sincerely thank all the reviewers for their time and effort in reviewing our manuscript and for providing valuable feedback. In response to their suggestions, we have conducted additional experiments and analyses. All newly added content can be found in the rebuttal file.

---

### Decision · Program_Chairs · 2024-09-04

**Decision:**

Accept

**Comment:**

Strengths:
- Insight that object level tokenization needs less data is useful
- Aligns text level representations like Llama with object level representations
- Long tail scenario is a data hungry regime, achieves few shot generalization

Weaknesses:
- Performance highly depends on pretrained tokenizer, and the latter needs lots of data. Inference latency might be slower given larger models?
- Trajectory end to end performance not better than PARA-driver. This is a major weakness and is not well explained.
- No experiments on hardware


Reviewer k2cG: Lack of explanation for MaIL's superior performance, its ability to handle long-horizon dependencies, and the relevance of Mamba's speed advantage in imitation learning. Also, the reviewer sought clarification on the emphasis on multi-modal input handling and the claim about MaIL's convergence with Transformers on large datasets.The authors provided motivations for MaIL's advantages, conducted experiments demonstrating its ability to handle long-horizon dependencies, and offered inference speed comparisons. They clarified their claims and conducted additional experiments with the full LIBERO dataset. They also justified their emphasis on MaIL's multi-modal handling capabilities.

Reviewer u9Yp: The paper could be misleading by claiming MaIL's outperformance without specifying the low-data setting. The ablation on dataset size was insufficient, and the Mamba Aggregation approach needed more elaboration. The authors acknowledged the potential misinterpretation and planned to correct it. They conducted additional experiments with the full dataset and provided a detailed explanation of the Mamba Aggregation approach.

Reviewer mQZR: The reviewer had concerns about the low success rates on the LIBERO benchmark, the definition of an epoch, whether the model fit the data well, and the absence of BC-RNN as a baseline. The authors attributed the low success rates to using a fraction of the dataset and provided results with the full dataset. They clarified the definition of an epoch, provided the number of gradient steps, conducted experiments with multiple checkpoints, and explained the exclusion of BC-RNN while providing results with an LSTM-based model.

Reviewer Vk9K: The reviewer suggested expanding the discussion of results, evaluating Mamba's ability to handle long-term context, and qualitatively analyzing state space representations. They also sought clarification on specific results and figure labels. The authors provided further discussion, conducted experiments with extended history lengths, and committed to visualizing state space representations. They also clarified figure labels and addressed specific questions about the results.